# Taste Masking of Promethazine Hydrochloride Using l-Arginine Polyamide-Based Nanocapsules

**DOI:** 10.3390/molecules28020748

**Published:** 2023-01-11

**Authors:** Hamad S. Alyami, Dalia Khalil Ali, Qais Jarrar, Abdolelah Jaradat, Hadeel Aburass, Abdul Aleem Mohammed, Mohammad H. Alyami, Alhassan H. Aodah, Eman Zmaily Dahmash

**Affiliations:** 1Department of Pharmaceutics, College of Pharmacy, Najran University, Najran 55461, Saudi Arabia; 2Department of Physiotherapy, Faculty of Allied Medical Sciences, Isra University, Amman 11622, Jordan; 3Department of Applied Pharmaceutical Sciences and Clinical Pharmacy, Faculty of Pharmacy, Isra University, Amman 11622, Jordan; 4National Center of Biotechnology, Life Science & Environment Research Institute, King Abdulaziz City for Science and Technology, Riyadh 11442, Saudi Arabia; 5Department of Chemical and Pharmaceutical Sciences, School of Life Sciences, Pharmacy and Chemistry, Kingston University London, Kingston upon Thames KT1 2EE, UK

**Keywords:** promethazine HCL, polyamide, taste masking, l-arginine, nanocapsules

## Abstract

Promethazine hydrochloride (PMZ), a potent H1-histamine blocker widely used to prevent motion sickness, dizziness, nausea, and vomiting, has a bitter taste. In the present study, taste masked PMZ nanocapsules (NCs) were prepared using an interfacial polycondensation technique. A one-step approach was used to expedite the synthesis of NCs made from a biocompatible and biodegradable polyamide based on l-arginine. The produced NCs had an average particle size of 193.63 ± 39.1 nm and a zeta potential of −31.7 ± 1.25 mV, indicating their stability. The NCs were characterized using differential scanning calorimetric analysis and X-ray diffraction, as well as transmission electron microscopy that demonstrated the formation of the NCs and the incorporation of PMZ within the polymer. The in vitro release study of the PMZ-loaded NCs displayed a 0.91 ± 0.02% release of PMZ after 10 min using artificial saliva as the dissolution media, indicating excellent taste masked particles. The in vivo study using mice revealed that the amount of fluid consumed by the PMZ-NCs group was significantly higher than that consumed by the free PMZ group (*p* < 0.05). This study confirmed that NCs using polyamides based on l-arginine and interfacial polycondensation can serve as a good platform for the effective taste masking of bitter actives.

## 1. Introduction

Patient compliance is critical for effective clinical outcomes and taste masking is key as it is positively related to patient compliance [1,2,3]. To enhance and even maintain patient compliance, particularly among pediatric patients, dosages need to be designed to deliver the required dose safely, reliably, and with acceptable taste [4]. Several dosage forms have been specifically developed for pediatric patients and/or geriatrics, including oral liquids (syrups, elixirs, and suspensions), mini-tablets, chewable tablets, and orally disintegrating tablets [5,6]. The World Health Organisation (WHO) has acknowledged the necessity of developing dosage forms that specifically target pediatrics. It specified that dosage forms that are targeted to pediatrics should be acceptable and palatable [7].

Taste masking entails the development of a strategy that delays or prevents the active pharmaceutical ingredient (API) from interacting with the taste buds, thus eliminating or decreasing the negative sensory response [8]. Several varied strategies have been reported for masking the undesirable taste of active pharmaceutical ingredients (APIs). The first generally used approach is based on physical masking using sweeteners and flavoring agents, which are commonly employed in the pharmaceutical industry; coating the final dosage form to prevent the immediate release of the API is another commonly used strategy. Further, other strategies, including the use of solid dispersion, the formation of salts or prodrugs, viscosity modifiers, or complexing, have been reported [6,8,9]. Nanotechnology has also been utilized for the entrapment of various drug types, such as the use of nanocrystals [10,11], solid lipid nanoparticles [12], nano-frameworks [13], PLGA nanoparticles [14], silica nanoparticles [15], chitosan nanoparticles [16], and liposomes [17]. 

Promethazine hydrochloride (PMZ) is a potent H_1_-histamine blocker that is widely used to prevent motion sickness, dizziness, nausea, and vomiting [3]. It is known for its bitter taste, which can make treatment adherence even more difficult [3]. Thus, taking the API in liquid form is not a practical option as it will provoke nausea and/or vomiting. Therefore, there is a need to produce a solid dosage form of PMZ that demonstrates no bitterness. PMZ is soluble in water and alcohol [18]. Several approaches have been reported in the literature regarding the taste masking of PMZ. One approach involves the use of a solid dispersion to mask the taste of PMZ [19]. Further, a study conducted by Kolhe et al. [20] employed solvent evaporation and a commonly used polymer (Eudragit E100) for surface coating. Extrusion was used to develop taste masked granules containing PMZ and Eudragit E100 that were further compressed into orally dissolving tablets [1]. Ganguly et al. [2] reported the use of an inclusion complex with β-Cyclodextrin to taste mask PMZ. 

Polyamides based on arginine amino acid (Arg-PA)s are biocompatible and biodegradable. Therefore, their use is generally associated with fewer adverse effects than other drug delivery systems [21,22]. The interfacial polycondensation method (IP) is a chemical process of nanoencapsulation that is used due to its high encapsulation efficiency. IP is built on the reaction of a hydrophilic monomer with a lipophilic monomer at the interface of the two immiscible solvents. Characteristically, in the synthesis of polyamide, the water phase contains the diamine and the organic phase consists of the diacid chloride and an organic solvent [23,24]. 

The interfacial polycondensation method, which is simple, offers the potential to encapsulate bitter APIs [16,17]. The added value for this process stems from using biocompatible and biodegradable materials for the development of polymers and the encapsulation of the bitter API using a one-step process. The synthesis of Arg-PA using IP has not been previously reported. Our previous work using arginine-based amino acid poly esteramide was used for the encapsulation of thymoquinone for pulmonary drug delivery [23]. We also investigated the use of IP to develop polyamides using cystine and tyrosine amino acids [24,25]. Furthermore, the application of amino-acid-based polymers and the IP process for taste masking was not reported. Therefore, this work aimed to synthesize an amino-acid-based polymer using l-arginine that encapsulates promethazine and facilitates suitable masking of the bitter taste of PMZ upon exposure to saliva. 

## 2. Results and Discussion

### 2.1. Synthesis of Polyamide Based on l-Arginine (Arg-PA) and Polyamide Based on l-Arginine Loaded with PMZ Nanocapsules (PMZ/Arg-PA NCs)

The Arg-PA was produced from the l-arginine amino acid, which acted as a diamine monomer to react with a 1,3-cyclohexane dicarbonyl chloride using the interfacial polymerization method, as shown in Figure 1. Amide bond formation reactions occurred at or near the organic–aqueous interface when the two solutions were mixed and vigorously stirred. The prepared Arg-PA contained MWt 97,300 g/mol. In addition, the prepared Arg-PA was soluble in methanol (CH_3_OH), Dimethylformamide (DMF), *N*-Methylpyrrolidone (NMP), and Dimethyl sulfoxide (DMSO), while it was insoluble in Tetrahydrofuran (THF), acetone, diethyl ether, and chloroform (CHCl_3_). 

Promethazine-loaded nanocapsules (PMZ/Arg-PA NCs) were created using simple, safe, and repeatable interfacial polycondensation. This condensation reaction occurs between the water phase containing the arginine amino acid and PMZ. The other phase consists of 1,3-cyclohexane dicarbonyl chloride in chloroform (CHCl_3_). The reaction required vigorous stirring to increase the surface area for the reaction, which facilitated the formation of fine PMZ/Arg-PA NCs. The polyamide condensation reaction occurred via the interaction between the amine group of the arginine and the carbonyl carbon on the carbonyl chloride at the interface of the chloroform–aqueous layers. Using this method, direct encapsulation of PMZ resulted in capsules with an average diameter of around 100 nm. 

Several formulations were investigated, as summarized in Table 1. Different concentrations of PMZ were investigated. Further, several washing processes were assessed to reduce surface-bound PMZ and enhance the effectiveness of the taste masking process. The results revealed that a high PMZ concentration did not lead to higher entrapment and/or encapsulation efficiencies; therefore, the medium level of PMZ (100 mg) was used for the rest of the studies (F6). This could be attributed to the capability of the polymer to encapsulate PMZ. Apparently, higher concentrations of PMZ did not effectively enhance the EE and EC. Additionally, increasing the organic solvent volume from 5 mL to 10 mL resulted in higher EE and EC (in F6), which could be attributed to the increased surface area between the chloroform–aqueous layers that enhanced the polymerization efficiency; hence, more PMZ can be incorporated within the produced polymer chains. Jyothi et al. reported that a change in the ratio of the Org: aqueous phase resulted in changes in the polymerization and encapsulation efficiency [26]. Furthermore, the produced polymer was not soluble in chloroform; therefore, an increasing organic phase volume may have resulted in an increase in polymer formation at the interface area between these two layers where the polymerization reaction took place. Similar results were reported in [27,28,29,30].

### 2.2. Fourier Transform Infrared (FTIR) Spectroscopy Analysis

The FTIR spectrum of PMZ showed a characteristic sharp band that represents the (C-N) bond at 1455 cm^−1^, and the band at 2380 cm^−1^ represents the (C-H) bond, as shown in Figure 2a. Similar results were reported previously [3,31]. The FTIR spectrum of Arg-PA revealed two carboxylic acid characteristic infrared stretching absorption bands. The C=O showed a stretching vibration at 1689 cm^−1^, and the O-H appeared as a broad band from 3319 to 2509 cm^−1^. On the other hand, a strong stretching vibration band for the formed amide group C=O was observed at around 1653 cm^−1^, and IR bands of the N-H bond appeared at 3160 cm^−1^, as shown in Figure 2b. The results of the l-arginine-based PA are reported in [23]. A comparison between PMZ/Arg-PA NCs and Arg-PA FTIR spectra shows the same characteristic bands (Figure 2b,c). This may be due to overlapping between the polymer and drug bands. The spectra exhibited characteristic bands corresponding to the different functional groups in the samples, as summarized in Table 2. Furthermore, PMZ is available as Promethazine hydrochloride, so the HCl band appeared at 2400 cm^−1^. When preparing the polymer and the formula, the PMZ and the l-arginine were dissolved in a solution of NaOH, which led to the disappearance of the HCl signal from Figure 2b,c.

### 2.3. Nuclear Magnetic Resonance (NMR) Spectroscopy

The synthesized Arg-PA polymer was characterized by ^1^H NMR and ^13^C-NMR spectroscopy. The spectra exhibited characteristic chemical shifts corresponding to the different atoms present in the samples, as shown in Figure 3a,b. The chemical shifts of the amide carbonyl carbon atoms in the Arg-PA are of particular importance because they highlighted the formation of the polyamide; the carbon signals of the C=O groups of the formed amide groups appeared between 169.2 and 168.3 ppm, as shown in Figure 3b. The spectra exhibit characteristic chemical shifts corresponding to the different atoms present in the samples ^1^H-NMR and ^13^C-NMR, and the chemical shifts for the various atoms, as summarized in Table 3.

### 2.4. Particle Size, Zeta Potential, and TEM Analysis

Table 4 illustrates the average hydrodynamic diameter, along with the polydispersity index (PDI) and zeta potential values of the PMZ particles, Arg-PA nanoparticles, and PMZ/Arg-PA NCs. The average hydrodynamic diameter of the nanoparticles was measured using the dynamic light scattering technique. For comparision, and to investigate the effect of the particle size of the drug upon the formulation, the hydrodynamic particle size of the PMZ was evalauted. Particles composed of the PMZ drug, as purchased from the supplier, possessed a large hydrodynamic diameter in the micrometre range (circa 1.5 µm), as displayed in Table 2. Although micronisation of the drug might reduce its dissolution in saliva, the drug is not encapsulated with a polymer in this case and the diffusion of the PMZ molecules into saliva is possible. On the other hand, the Arg-PA nanoparticles exhibited a nanometer-size range (~189 nm) that is within the optimum range for drug delivery [32]. The average diameter of Arg-PA was slightly increased to 193 nm after PMZ incorporation, which is expected due to drug entrapment [16,17]. Furthermore, the zeta potential values were measured for the particles, and the PMZ particles displayed a positive charge value (+8.4), as depicted in Table 4. The positive charge of the promethazine particles is attributed to the protonated tertiary amine group of PMZ ((-RNH^+^(CH_3_)_2_)) [18]. Additionally, the arginine nanoparticles displayed a neutral to slightly negative charge (−0.5), which is thought to be due to the presence of a negatively charged carboxylate-containing pendant group in the polymer chains. Surprisingly, the addition of basic PMZ to Arg-PA nanoparticles during drug entrapment caused a drastic increase in the particles’ negative charge (−31.7). The absolute value of the zeta potential of drug-loaded nanoparticles was greater than 25, which is necessary to maintain the colloidal stability of the particles [33]. 

Moreover, in images of drug-loaded nanocapsules obtained using a transmission electron microscope (TEM), the particles show a spherical shape, as demonstrated by the TEM micrographs depicted in Figure 4a,b. Moreover, the exact particle diameter of the drug-loaded nanoparticles was calculated from the TEM micrographs using ImageJ (Fiji) software, and the frequency distribution for these data is depicted in Figure 4c. The particles demonstrated an average diameter of around 100 nm, which is less than that measured by the dynamic light scattering technique (which was around 200 nm). This discrepancy has been previously reported by many researchers as the diffraction laser technique measures the average diameter of the nanoparticles, including the hydration shell surrounding the NPs’ surfaces, while the TEM image shows the exact diameter of the NPs [34,35,36]. However, in this case, there was a dramatic difference between the particle diameter measured by dynamic light scattering and that measured by TEM; this difference could be ascribed to the formation of the particles’ dimers (aggregates) in the nanosuspension that led to an increased particle diameter of nearly two-fold more than the actual diameter.

### 2.5. Differential Scanning Calorimetry (DSC) Analysis

Figure 5 displays the DSC graphs for the PMZ drug alone (a), Arg-PA (b), the physical mixture of PMZ and Arg-PA (c), and PMZ/Arg-PA NCs (d). There is an endothermic peak at 233 °C, indicating the melting point of promethazine [37,38], as demonstrated in Figure 5a. Figure 5b shows a shallow peak at around 43.47 °C, which could be attributed to the glass transition exhibited by Arg-PA as most polymers are amorphous and demonstrate a glass transition point. It is noteworthy that Arg-PA has been synthesised for the first time here; therefore, its glass transition temperature was not previously reported in the literature. From Figure 5c, it can be seen that the physical mixture composed of PMZ and Arg-PA affected the melting point of promethazine, with a dramatic reduction in the latent heat of fusion (enthalpy); however, the drug still melted at the same temperature point (i.e., around 225 °C). It is also worth noting that although the drug retained its melting point, surprisingly, the polymer did not display a definitive glass transition point in the recorded temperature range, it only displayed a very shallow and slightly broadened endothermic peak at around 25 °C. This could suggest that the physical mixture may have created some weak yet significant intermolecular interactions between PMZ and the polymer, such as hydrogen bonds between the polymer carboxylic acid groups (RCOOH) and the tertiary amine groups of the drug (R3N) that affected the polymer Tg. It is thought that the inclusion of the drug within the polymer matrix could have initiated weak intermolecular interactions with long distance, which might have increased the associated free volume of the polymer chains by separating them apart from each other. Accordingly, the polymer chains were able to move freely with high flexibility, resulting in a dramatic drop in the glass transition of the polymer probably below the measurement range (below 29 °C). This phenomenon has been widely described in the literature; for instance, the inclusion of glycerol molecules between the polymer chains usually exerts a plasticising effect and a significant reduction in the polymer glass transition due to the formation of the hydrogen bonds and an increase in the free associated volume of the polymer chains [39,40,41,42,43]. 

Overall, these observations could indicate that mixing the drug with the polymer might have influenced the degree of drug crystallinity; however, the interaction was not able to completely eliminate the endothermic peak of the drug at its melting point. In contrast, Figure 5d shows that the drug melting point completely disappeared, meaning that the drug was converted from a crystalline to an amorphous structure upon drug encapsulation by the polymer. This indicates a strong interaction between Arg-PA and PMZ as a result of the polymer-drug or the association or incorporation of the drug into Arg-PA NPs. Possible intermolecular interactions include an ion–dipole interaction between the protonated nitrogen in the promethazine drug (-RNH^+^(CH_3_)_2_) and the carbonyl oxygen (R-C=O) in polyarginine. Another possible interaction is the ion–dipole interaction between the guanidinium group (NH_2_-C-N=N-R) in the polyarginine and sulfur (R-S-R) in promethazine. This could have led to the disruption of the crystal lattice arrangement of the drug, which could explain the loss of drug crystallinity, as indicated by the disappearance of the drug melting point. It should also be noted that the glass transition temperature of the polymer, as shown in Figure 5b, has not been clearly observed as the usual small endothermic peak, but rather, an extremely shallow and highly broadened peak has appeared, which started at around 25 °C. This could be attributed to all of the possible interactions mentioned above between the drug and polymer functional groups, as they may have shifted the Tg to a lower temperature point outside of the measured temperature range (below 29 °C). Consequently, these possible interactions indicate the successful incorporation of PMZ into Arg-PA nanoparticles. Similar results were previously reported, where the drug endothermic peak dissapeared [38,44,45].

### 2.6. X-ray Diffraction Analysis

Figure 6 demonstrates the X-ray diffractograms of the PMZ alone (a), Arg-PA alone (b), the physical mixture of Arg-PA and PMZ (d), and PMZ/Arg-PA NCs (d). It is clear from Figure 6a that the drug has a crystalline structure, as indicated by the presence of well-defined and pronounced Bragg peaks. On the other hand, well-defined peaks are barely observable for the polymer in Figure 6b, with a broad peak appearing at around 5° instead of a sharp peak, which usually appears for the crystalline form and reinforces the amorphous structure of Arg-PA. However, it should be noted that the polymer exhibited a slight degree of crystallinity as small sharp peaks appeared at around 32° and 45°. Similar results were reported for the slight crystalinity of l-arginine based polymers [23]. It is clear from Figure 6c that the superposition of both the drug and polymer peaks was obtained upon physically mixing the polymer and the drug. This indicates that the presence of the polymer with an amorphous structure did not affect the lattice distances of the drug molecules. In contrast, Figure 6d reveals that the drug peaks disappeared due to drug encapsulation within the matrix of Arg-PA NCs. This also confirms the presence of a strong interaction between the components of the sample, i.e., the PMZ drug and Arg-PA, which suggests that the polymer affected the arrangement of the drug upon its incorporation into the nanoparticles [46]. Another possible explanation is that due to the entrapment of the drug into the nanoparticles, the X-ray was not able to penetrate through the NPs; hence, the XRD pattern for the drug was not recorded [47]. Previous studies reported that the encapsulation of a PMZ transformed it from a crystalline to an amorphous structure [1]. 

### 2.7. HPLC Method for the Quantification of Promethazine 

An HPLC method for the quantification of PMZ was developed and validated according to the ICH guidelines for analytical method validation [48]. The retention time for PMZ was 5.1 ± 0.12 min and the polymer components did not interfere with the PMZ peak. Table 5 summarizes all validation parameters, demonstrating a suitable and reproducible method. Representative chromatograms are presented in Appendix A.

### 2.8. In Vitro Assessment of Taste Masking 

Four formulas were evaluated for taste masking. A small volume of artificial saliva was utilized to assess the amount of PMZ released immediately from the formula, the results of which are depicted in Table 6. Formulation with a low drug concentration (F1) resulted in a low release of 7.41% (less than 0.4 mg of PMZ); however, higher doses were investigated to better match the dosage of marketed products (25 mg and 50 mg). Although double washing of the nanoparticles resulted in a reduction in EE and EC, it resulted in better taste masking, as evident from the low release after 5 min and 10 min of F3 when compared to F2. Interestingly, however, higher doses with three rounds of washing, as in F6, where the organic phase volume was increased, resulted in higher EE and EC and better taste masking. In this instance, as little as 1% of PMZ was released over 10 min, which is more than sufficient for the drug to mix with the buccal cavity and produce a bitter taste. A study by Haware et al. reported the bitterness threshold of PMZ to be 0.1 mg/10 mL (10 µg/mL) in 30 s [1]. In comparison, after 1 min, the optimal formula (F6) did not release any PMZ; therefore, F6 did not reach the bitterness threshold. Similarly, formulations F1 and F3 did not exceed the bitterness threshold after 1 min; however, F1 only contained 5.02 mg of PMZ in a 200 mg formula.

### 2.9. In Vivo Assessment of Taste Masking 

Data obtained from the animal study are presented in Figure 7. The comparative analysis revealed that mice exposed to free- and polymer-loaded drugs consumed significantly lower amounts of fluids than the control mice did. However, the amount of fluid consumed by the polymer-PMZ group was significantly higher than that consumed by the free PMZ group (*p* < 0.05).

Mice exhibit a natural aversion toward bitter substances that represent a potential source of numerous toxins and harmful chemicals [49]. According to recent research, mice have a comparable tasting system to humans in that they can recognize bitter flavors [50]. It has been reported that the genome of mice consists of pairs of genes that regulate the biosynthesis and functions of bitter taste receptors [51]. Laboratory mice have been frequently used as a sensitive model for testing the palatability of bitter substances in preclinical investigations [52,53]. The findings of the present study revealed that BALB/c mice showed a marked aversion towards the uncoated form of PMZ, which was determined by a significant decrease in the amount of the drug consumed. On the other hand, mice exposed to the encapsulated form of PMZ exhibited a significant decrease in aversion, indicating that the proposed drug encapsulation system in this study increased drug palatability through a mechanism that may be related to its ability to delay the onset of drug release. The findings of the in vivo evaluation of the optimal formula (F6) were compatible with the in vitro drug release profile, which demonstrated a low release rate (0.91 ± 0.02%) over 10 min, allowing the mice to consume more fluids in the early stage of the experiment.

### 2.10. Release Study

According to the WHO, dosage forms that specifically target pediatrics need to be acceptable and palatable [7]. Both the in vitro and in vivo tests demonstrated the enhanced palatability of PMZ upon encapsulation. However, the produced NCs need to demonstrate the ability to release the encapsulated drug to produce its acceptable and required effect. The results of the release study are depicted in Figure 8. Despite the presence of a large volume of the dissolution medium, the particles did not demonstrate a burst effect. This is attributed to the washing process that removed all PMZ that might be attracted to the surface of the polymer. A lag time of around 15 min was needed for the polymer to start degrading and for it to release the water-soluble API. The extended release of PMZ is attributed to the entrapment of PMZ within the biodegradable polymer. Complete release of PMZ was achieved within 4 h. 

The release pattern was fitted using various models according to [23,54,55]. The release profile best fits the Higuchi model (see Table 7, and Appendix A). This model refers to the diffusion of PMZ from the polymer. To better understand the diffusion pattern of PMZ, the Korsmeyer–Peppas model was fitted and it revealed an excellent correlation with an *n* value of 2.46, indicating a super case II transport relaxation model of spherical particles [56]. In this model, the PMZ release is controlled mainly by the relaxation of the polymer chains. Despite the high solubility of PMZ, its release requires the polymeric chain to relax and degrade for the effective dissolution of the drug [57]. This is favorable because a delay of drug release is required to prevent the bitter tasting drug reaching the buccal cavity during taste masking processes. 

## 3. Materials and Methods

### 3.1. Material

Promethazine hydrochloride (PMZ) was obtained from Carbosynth Limited (Compton, UK). Sodium chloride (NaCl), sodium hydroxide (NaOH), 1,3-cyclohexane dicarboxylic chloride, disodium hydrogen phosphate, and potassium dihydrogen phosphate were purchased from AZ Chemicals, Inc. (Thunder Bay, ON, Canada). l-arginine was obtained from Acros Organics (Geel, Belgium). Furthermore, HPLC-grade distilled water, DMSO, acetonitrile, methanol, chloroform, phosphoric acid, and trifluoroacetic acid (TFA) were purchased from Alpha Chemika (Mumbai, India).

### 3.2. Methods

#### 3.2.1. Synthesis of Polyamide Based on l-Arginine (Arg-PA)

A mixture of l-arginine (0.52 g, 2.5 mmol), NaOH (0.20 g, 5.0 mmol) and 1 g of NaCl was dissolved in 5 mL of distilled water and placed in a 50 mL round-bottomed flask. The resulting aqueous solution was cooled to 0–5 °C. Next, 1,3-cyclohexane dicarboxylic chloride (2.5 mmol, 0.50 g) was dissolved in 5 mL of chloroform (CHCl_3_), added dropwise to the prepared arginine containing the aqueous solution, and stirred at 1500 rpm for 60 min. A pale-yellow precipitate [arginine polyamide (Arg-PA)] was formed and collected by performing suction filtration and washed three times with distilled water. Finally, the Arg-PA was dried in a freeze-dryer until completely dry.

#### 3.2.2. Synthesis of Polyamide Nanocapsules Loaded with PMZ (PMZ/Arg-PA) NCs

A mixture of l-arginine (0.52 g, 2.5 mmol) and NaOH (0.20 g, 5.0 mmol) and 1 g of NaCl, which were dissolved in 5 mL of distilled water, was placed in a 50 mL round-bottomed flask; afterwards, different amounts of PMZ (10 mg, 100 mg, 200 mg, or 500 mg) were dissolved in a mixture of 10 mL of acetone and 10 mL of distilled water. The resulting aqueous solution was cooled to 0–5 °C. Next, 1,3-cyclohexane dicarboxylic chloride (2.5 mmol, 0.50 g) was dissolved in 5 mL of chloroform (CHCl3) and added dropwise to the prepared arginine–PMZ aqueous solution, which was stirred at 1500 rpm for 60 min. The formation of a pale-yellow precipitate (PMZ/Arg-PA) NC was collected by performing suction filtration and washed three times with distilled water. Finally, the (PMZ/Arg-PA) NC was dried in a freeze-dryer until completely dry.

#### 3.2.3. Molecular Weight Measurement

The molecular weight of the produced polymer was measured in an aqueous suspension by performing dynamic light scattering using a Nicomp N3000 (Billerica, MA, USA). 

#### 3.2.4. Entrapment and Encapsulation Efficiency 

The amount of loaded PMZ is expressed as the entrapment (EE) and encapsulation (EC) efficiencies within the nanoparticles. EE was calculated using the following equation: (1)$$\mathrm{EE}\;\left(\%\right)=\frac{{\mathrm{PMZ}}_{t}-{\mathrm{PMZ}}_{s}}{{\mathrm{PMZ}}_{t}}\times 100$$
where the PMZ_t_ is the total amount of PMZ that was used to prepare the formula, while PMZ_s_ is the amount of PMZ that is present in the supernatant (which is not entrapped within the nanoparticles). The EC was calculated based on the content of PMZ in the final formula. About 50 mg of the final dried formula was accurately measured and dissolved in 50 mL of acetone, filtered using a syringe filter with 0.45 µm apertures, and assayed using the HPLC method. 

#### 3.2.5. Quantification of PMZ Using HPLC

PMZ quantification was conducted using the Dionex Softron HPLC system (Thermo Fisher Scientific Inc., Waltham, MA, USA) coupled with a gradient pump and UV detector. The samples were run in the HPLC using a reversed-phase C18 Fortis column (4.6 × 250 mm, 5 μm) (Fortis Technologies Ltd., Neston, UK). A mobile phase consisting of A: methanol and B: 0.025% trifluoroacetic acid (TFA) with a pH 2.6. mobile phase elution was isocratic at 70:30 (*v*/*v*) of A: B. For column equilibration, an isocratic elution was applied at 100:00 for 10 min to prepare the column for the next sample. The UV detector was set at 249 nm. The temperature was set at 25 °C, and the flow rate was set at 1 mL/min, while the injection volume of each sample was 20 µL. Different standards of different concentrations ranging from 7.813 to 125 µg/mL were used to construct the calibration curve. The retention time of the promethazine drug was 5.1 min. The analytical method was validated according to the International Conference of Harmonisation (ICH) guidelines in terms of specificity, selectivity, linearity, the limit of detection (LOD), the limit of quantification (LOQ), and reproducibility [48]. 

#### 3.2.6. Fourier Transform Infrared Spectroscopy (FTIR) Analysis

The FTIR spectra of l-Arginine-based polymer and the PMZ/Arg-PA NCs were obtained using a PerkinElmer FTIR spectrometer (Akron, OH, USA). The FTIR spectra were further analyzed using SpectrumTM 10 software obtained from PerkinElmer (OH, USA). A few milligrams were positioned on the main lens, the FTIR scans were recorded over the range of 500–4000 cm^−1^, and the resolution was set at 2 cm^−1^.

#### 3.2.7. Nuclear Magnetic Resonance Spectroscopy (NMR) Analysis 

The 1H NMR at 500 MHz and the 13C NMR at 125 MHz analysis of the Arg-PA NCs were conducted using a Bruker Avance DPX NMR spectrometer (Bruker DPX-500) from Boston, MA, USA. Tetramethylsilane (TMS) was employed as the internal standard. Chemical shifts were described as parts per million (ppm). Splitting was recorded as follows: “s” for singlet, “d” for doublet, “t” for triplet, “q” for the quartet, and “m” for multiplets. 

#### 3.2.8. Particle Size Analysis (PSA)/Measurement

The average particle size, polydispersity (PDI), and zeta potential of the synthesized nanocapsules were measured using the dynamic light scattering technique (Malvern Zetasizer Nano ZS90) with the Malvern Instrument, Malvern, UK. Around 5 mg of Arg-PA NCs and PMZ-loaded Arg-PA NCs were suspended in 1 mL deionized water containing 10 mM NaCl. The samples were then diluted to produce a concentration of 0.1 mg/mL. The temperature was set to 20 °C and the viscosity of the dispersant was 1.2 Cp with a 1.362 refractive index.

#### 3.2.9. X-ray Diffraction Studies (XRD)

XRD analyses were performed using Rigaku MiniFlex 300/600 (Tokyo, Japan), equipped with a Cu Ka radiation source excitation voltage of 40 kV, and 15 mA current. The data were recorded between 2θ, ranging from 0° to 90° at a scan speed of 10 °/min. All samples were placed on a glass holder and scanned in triplicate. OriginPro^®^ software was employed to analyze the produced scans from the OriginLab Corporation (Northampton, MA, USA).

#### 3.2.10. Transmission Electron Microscopy (TEM) Analysis 

FEI Morgagni 268D TEM (Hillsboro, OR, USA) was used for the imaging of the PMZ-loaded Arg-PA NCs as follows: a 0.05 mg/mL suspension of drug-loaded PMZ/Arg-PA NCs was prepared in 1 mL deionized water. Then, 15 μL of the prepared suspension was dropped onto a copper grid and left for 10 min to allow the nanoparticles to settle on the grid. Finally, the solvent was removed with filter paper and the sample was loaded into TEM for imaging. All acquired images were further processed using ImageJ software (Fiji) version 1.53t.

#### 3.2.11. Differential Scanning Calorimetry Analysis (DSC)

DSC analysis of the synthesized Arg-PA and PMZ/Arg-PA NCs was achieved using a DSC Q200-TA instrument (USA). The process was conducted by placing 2–3 mg of the samples in an aluminum pan and heating them at a rate of 10 °C per minute until they reached 300 °C. Nitrogen gas was purged continuously at a rate of 50 mL/min.

#### 3.2.12. In Vitro Assessment of Taste Masking Using a Release Study 

The release of PMZ was carried out using a dissolution test of all prepared formulations in artificial saliva (AS), which was composed of the following: disodium hydrogen phosphate (0.2382%), potassium dihydrogen phosphate (0.019%), and sodium chloride (0.8%). Then, the pH of the AS was adjusted to 6.75 using phosphoric acid. After that, 200 mg of each formulation consisting of PMZ/Arg-PA NCs was placed in 20 mL of AS in a beaker, which was stirred at 120 rpm, and the temperature was set at 37 °C. A sample of 1.5 mL was taken at specified intervals (0, 1, 2, 3, 4, 5, 6, 7, 8, 9, and 10 min) and was replaced by 1.5 mL of AS immediately to maintain sink conditions. All samples were filtered using a 0.45 µm syringe filter, then analyzed using HPLC to determine the quantity of PMZ released from each formulation.

#### 3.2.13. Release Study 

The release of PMZ from the PMZ/Arg-PA NCs was carried out using the USP I dissolution apparatus (Hanson Research SR6). Around 100 mg of the formula (containing around 26 mg of PMZ) was accurately weighed and placed in the dissolution vessel containing 800 mL of distilled water. The rotation speed was set at 50 rpm at a temperature of 37 ± 0.5° C. The samples were withdrawn (2 mL) at the following time intervals: 15, 30, 45, and 60 min, and then at 2, 3, and 4 h. The samples were filtered using a 0.45 µm syringe filter before determining the amount of dissolved drug using the HPLC method.

The release profile of the optimal formulation (F6) was evaluated using the commonly employed kinetic mathematical models according to the following Equations (2)–(6): (2)$$\mathrm{Zero}\mathrm{Order}:{\mathrm{PMZ}}_{t}={\mathrm{PMZ}}_{0}+K_{0}\times t$$
(3)$$\mathrm{First}\mathrm{Order}:\log{\mathrm{PMZ}}_{t}=\log{\mathrm{PMZ}}_{0}-K_{1}\times 2.303t$$
(4)Higuchi:PMZt=Kh×t12
(5)$$\mathrm{Hixon}\mathrm{Crowell}:\sqrt[{3}]{{\mathrm{PMZ}}_{\mathrm{rt}}}=k.t$$
(6)$$\mathrm{Korsmeyer}–\mathrm{Peppas}:\frac{{\mathrm{PMZ}}_{t}}{{\mathrm{PMZ}}_{\infty }}=K_{p}\times t^{n}$$
where PMZ_t_ is the quantity of PMZ released at time t, PMZ_0_ is the initial quantity of PMZ, K_0_ is the zero-order rate constant, t is time, K_1_ is the first order rate constant, K_h_ represents the Higuchi rate constant, PMZ_rt_ represented the remaining quantity of PMZ, k is the Keppa constant, PMZ_∞_ denotes the total quantity of PMZ to be released, and K_p_ and n are the release constant and exponent, respectively. (To calculate the n value, the log percentage released versus log time is plotted and n is the slope of the produced line) [23].

#### 3.2.14. Animal Study

##### Animal Husbandry and Care

A total of 63 male mice of the same age (8–12 weeks old) were used in this study. These mice were housed and used according to the guidelines of the Institution of Animal Care and Use Committee (IACUC) [58]. The mice were kept under normal conditions (room temperature 22–24 °C and at a relative humidity of 50–60%) and acclimatized for seven days before any animal experiments. The use of animals in this study was approved by the Research Ethics Committee at Al-Isra University, Jordan (SREC/22/5/24 in April 2022).

##### Palatability Assessment

A simple method was developed to assess the taste acceptability and palatability of PMZ-loaded Arg-PA NCs in comparison to free PMZ in mice. Three groups of mice were used (seven mice were assigned to each group and each group was repeated three times). These mice were housed at room temperature (22–24 °C) and fasted overnight. Following a 17 h fast, the mice were provided free access to a drinking source, as follows: Control group: the mice were provided with distilled water.Free drug group: the mice in this group were provided with PMZ suspended in distilled water at 1.5 mg/mL.Polymer–drug group: the mice in this group were provided with PMZ/Arg-PA NCs suspended in distilled water at 6 mg/mL (the PMZ content was 1.56 mg/mL).

The amount of fluid consumed over 10 min was measured and used as an index of palatability. The test was repeated in triplicate to ensure the reproducibility of the results. Data were presented as the mean (of triplicate trials) ± standard deviation (SD). Significant differences between the groups were determined using a one-way analysis of variance (ANOVA) and Tukey’s post-test. A probability value of less than 0.05 was considered statistically significant.

#### 3.2.15. Statistical Analysis 

The one-way ANOVA and the Tukey post-test for statistical analysis were carried out using Minitab version 18. All data were expressed as the mean ± SD or relative standard deviation (RSD) and *p* < 0.05 was considered to be the level of significance.

## 4. Conclusions

In the present study, taste masked PMZ nanocapsules, which were encapsulated with a polymer made of a polyamide based on l-arginine nanocapsules, were successfully prepared using the interfacial polycondensation method. The morphological, molecular profiling, in vitro, and in-vivo results of the nanocapsules support their suitability for taste masking. The TEM analysis showed spherical PMZ/Arg-PA NCs with an average particle size of 100 nm. Increasing the organic phase ratio with three washing cycles resulted in high entrapment and encapsulation efficiencies (93.5% and 26.58%, respectively). Both FTIR and NMR analyses confirmed the formation of the polymer. The in vitro assessment of the NC revealed that the produced NCs delivered less than 1% PMZ over 10 min in artificial saliva, whereas the in vivo results confirmed the superiority of the produced NP over free PMZ in liquid consumption (*p* < 0.05). The employed method facilitated the production of a simple cost-effective process for taste masking water-soluble bitter actives. 

## Figures and Tables

**Figure 1 molecules-28-00748-f001:**
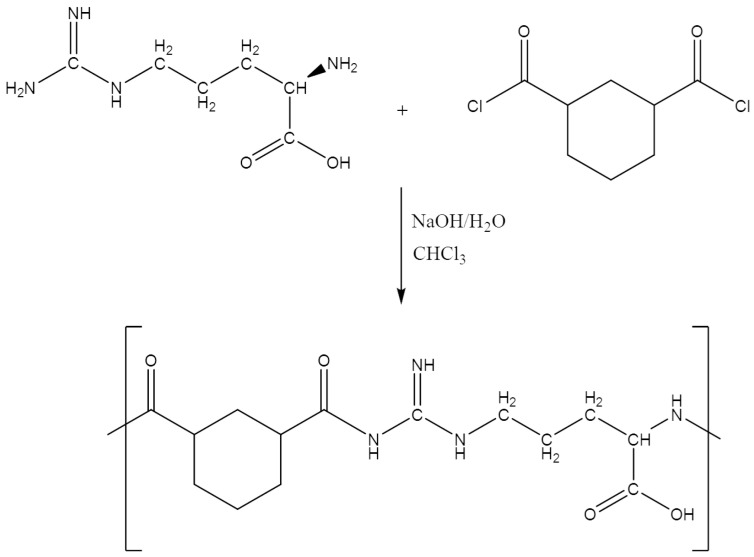
Synthesis of the polyamide by interfacial polycondensation of l-arginine amino acid and cis/trans-1,3-cyclohexane dicarbonyl chloride.

**Figure 2 molecules-28-00748-f002:**
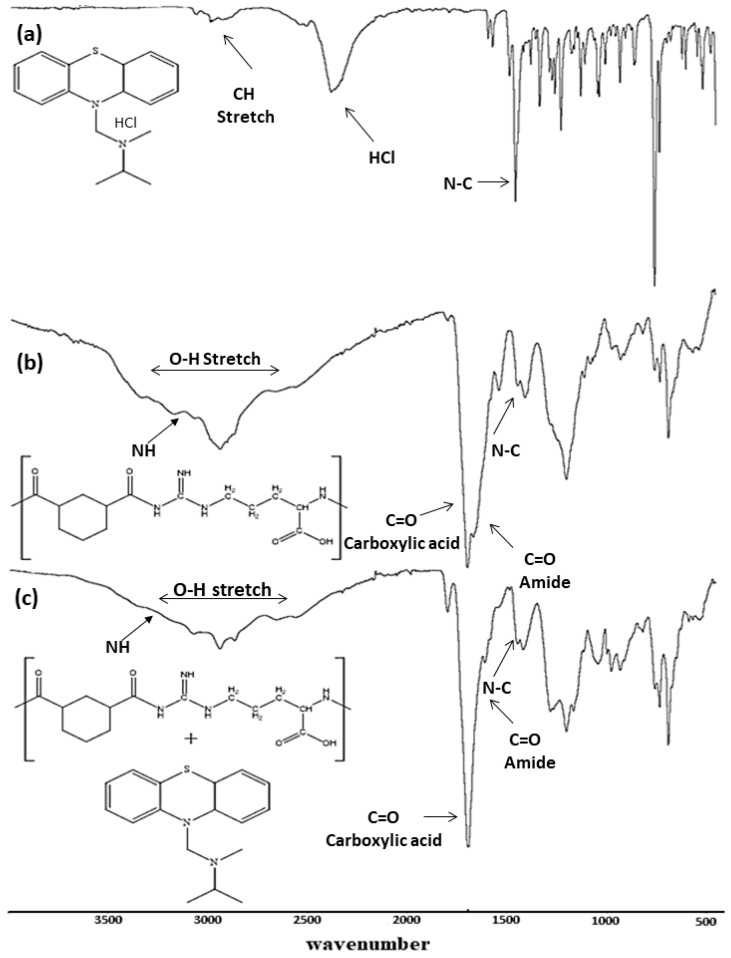
FTIR spectra of (**a**) PMZ drug, (**b**)Arg-PA, and (**c**) PMZ/Arg-PA NCs.

**Figure 3 molecules-28-00748-f003:**
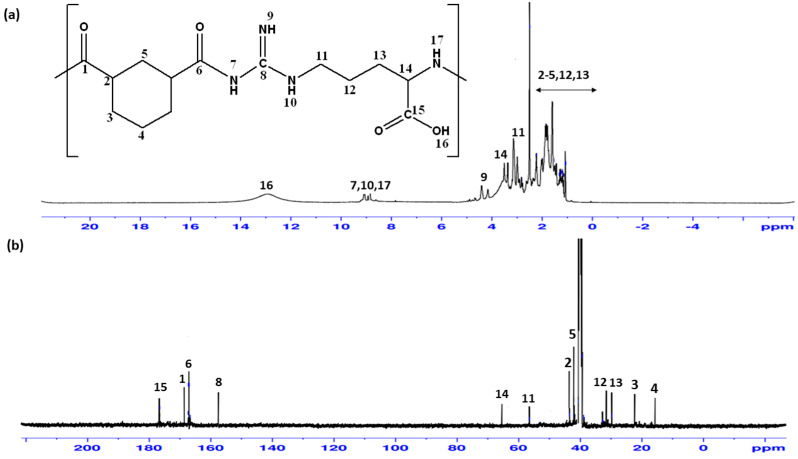
NMR spectra of Arg-PA: (**a**) ^1^H NMR, (**b**) ^13^C NMR.

**Figure 4 molecules-28-00748-f004:**
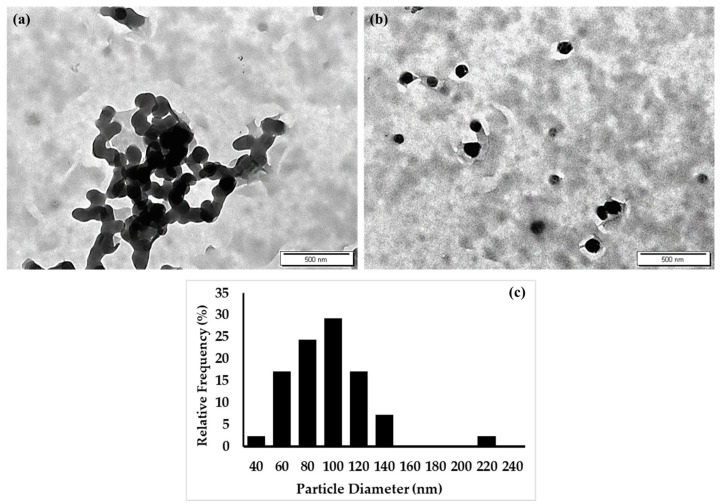
(**a**,**b**) TEM micrographs of PMZ/Arg-PA NCs highlighting the particle shape and size. Scale bar = 500 nm. (**c**) Histograms of the frequency distribution representing the exact diameters of PMZ/Arg-PA NCs and the corresponding percentages/proportions of each diameter in the sample; the data were generated from TEM images of drug-loaded Arg-PA nanoparticles containing several particles (*n* = 40).

**Figure 5 molecules-28-00748-f005:**
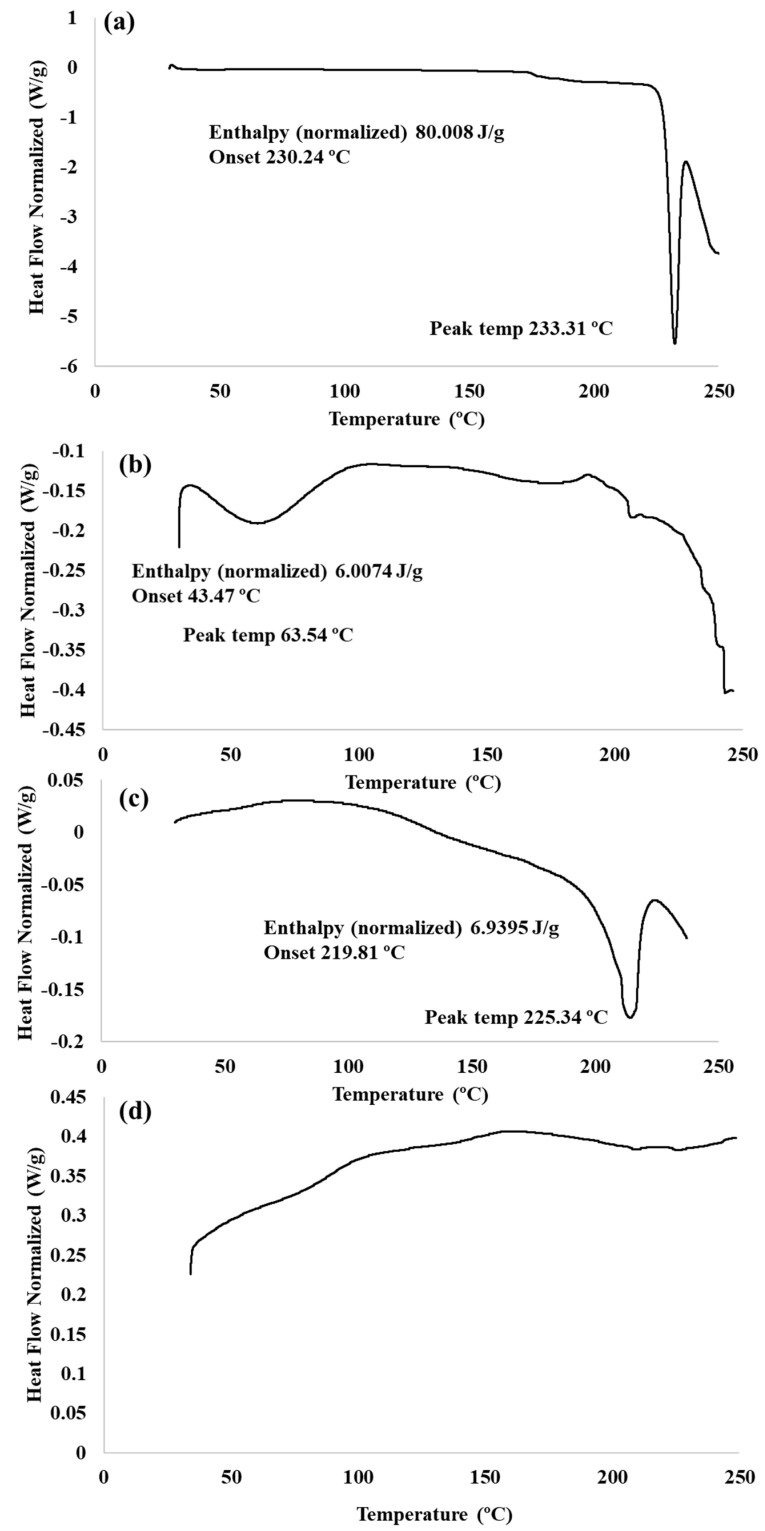
DSC curves for (**a**) PMZ, (**b**) Arg-PA, (**c**) the PMZ-Arg-PA physical mixture, and (**d**) PMZ/Arg-PA NCs.

**Figure 6 molecules-28-00748-f006:**
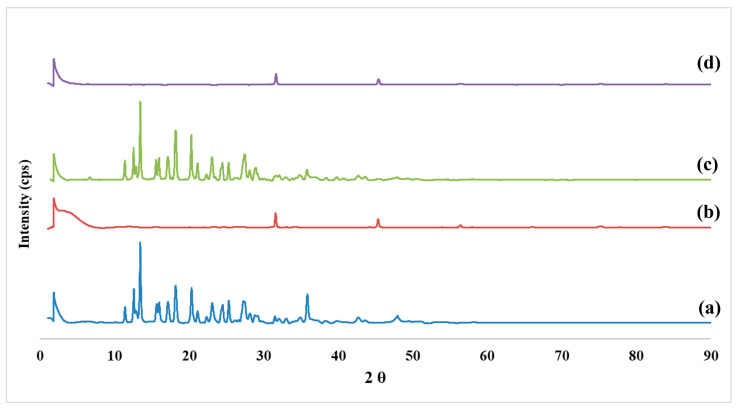
XRD pattern of (**a**) PMZ, (**b**) Arg-PA, (**c**) the PMZ- Arg-PA physical mixture, and (**d**) PMZ-Arg-PA NCs.

**Figure 7 molecules-28-00748-f007:**
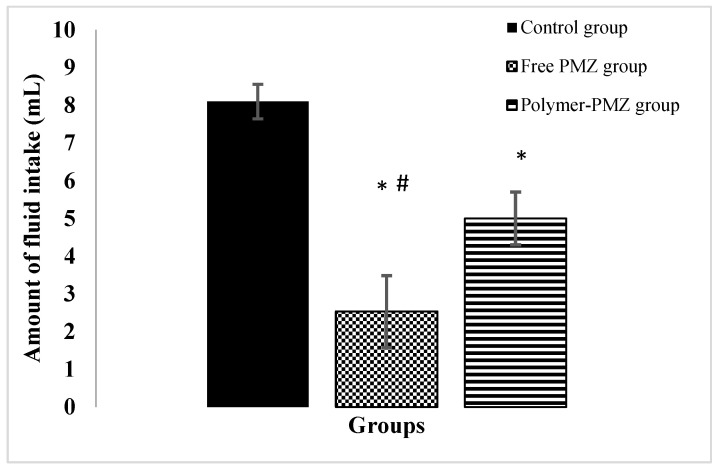
Fluid intake in various experimental groups. (*) indicates a significant difference (*p* < 0.05) from the control group according to the Tukey HSD test. (#) indicates a significant difference (*p* < 0.05) between the free PMZ group and the polymer-PMZ group according to the Tukey HSD test.

**Figure 8 molecules-28-00748-f008:**
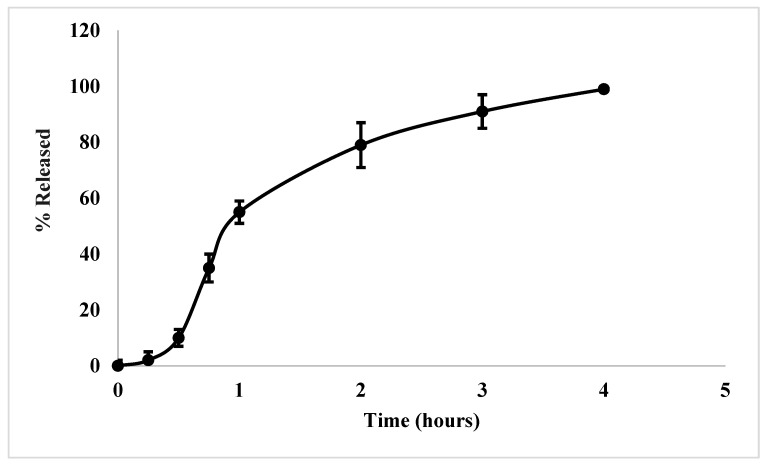
The release profile of PMZ from PMZ/Arg-PA NCs formula F6 over 4 h (mean ± SD, *n* = 3).

**Table 1 molecules-28-00748-t001:** Summary of the composition and evaluation parameters of formulas used in developing the taste masked nanoparticles.

Formula	Amount of PMZ (mg)	Organic: Aqueous Medium Volume (mL)	Monomers (l-Arginine: 1,3-Cyclohexane Dicarboxylic Chloride) (g)	Number of Washing Cycles	EE * (%)	EC ** (%)
F1	10	5: 25	0.52: 0.5	1	29.10	2.51
F2	100	5: 25	0.52: 0.5	1	24.55	13.21
F3	100	5: 25	0.52: 0.5	2	19.89	11.05
F4	200	5: 25	0.52: 0.5	2	18.55	12.22
F5	500	5: 25	0.52: 0.5	2	8.54	9.87
F6	100	10: 25	0.52: 0.5	3	93.5	26.58

* EE: entrapment efficiency, ** EC: encapsulation efficiency.

**Table 2 molecules-28-00748-t002:** FTIR spectral properties of PMZ, Arg-PA, PMZ/Arg-PA.

	OH Amide (cm^−1^) Stretch	CH (cm^−1^) Aromatic	CH (cm^−1^) Aliphatic	NH Amide (cm^−1^) Stretch	-C-N (cm^−1^)	-COOH (cm^−1^)	-CON (cm^−1^)
PMZ		2985	2893	-	1455	1699	-
Arg-PA	3332–2529	2995	2912	3181	1449	1711	1671
PMZ/Arg-PA	3329–2541	2986	2913	3131	1446	1702	1669

**Table 3 molecules-28-00748-t003:** ^1^H NMR and ^13^C NMR data of the synthesized Arg-PA.

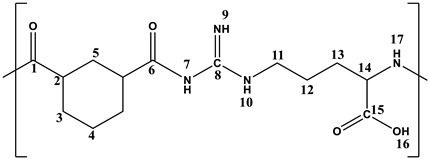																	
Chemical Shift (ppm)																	
	1	2	3	4	5	6	7	8	9	10	11	12	13	14	15	16	17
**^1^H**	-	2.20-1.10				-	8.86	-	4.32	9.41	3.11	2.20-1.10		3.61	-	13.11	9.01
**^13^C**	169.2	43.4	22.2	15.6	42.0	168.3	-	157.5	-	-	56.5	32.7	29.7	65.4	176.7	2-	-

**Table 4 molecules-28-00748-t004:** Particle size analysis of promethazine, the polymer, and promethazine-polymer-loaded nanoparticles (F6) (PMZ-POL NP), (mean ± SD, *n* = 3).

Material	Particle Size (nm) Mean ± SD, *n* = 3	PDA * Mean ± SD, *n* = 3	Zeta Potential (mV) Mean ± SD, *n* = 3
PMZ	1587 ± 262.75	0.467 ± 0.06	8.41 ± 0.78
Arg-PA	189.33 ± 29.69	0.26 ± 0.03	−0.53 ± 0.95
PMZ/Arg-PA NCs	193.63 ± 39.1	0.42 ± 0.05	−31.7 ± 1.25

* PDA: polydispersity index.

**Table 5 molecules-28-00748-t005:** Validation parameters of PMZ using the HPLC method.

Regression Equation (Calibration Curve)	Area under the Curve = 0.6267X (Concentration) − 2.4862 (R^2^: 0.9992)	
Limit of detection (LOD) µg/mL	4.06	
Limit of quantification (LOQ) µg/ml	12.31	
Range (µg/mL)	7.813–125	
	Intraday% recovery (mean ± SD) (*n* = 3)	Interday% recovery (mean ± SD) (*n* = 9)
125 µg/mL	98.64 ± 4.18	97.94 ± 5.74
31.25 µg/mL	100.95 ± 6.15	99.07 ± 3.12
3.9 µg/mL	99.75 ± 4.22	101.25 ± 4.46

Precision (concentration 31.25 µg/mL) (mean ± SD, RSD) (*n* = 10)	99.69 ± 2.25, 1.98%	

**Table 6 molecules-28-00748-t006:** Summary of the percentage cumulative release of PMZ from various formulas (200 mg of each formula added to 20 mL of artificial saliva); results are presented as the mean ± SD, *n* = 3.

% Released [PMZ Concentration (µg/mL)]				
Time (Minutes)	F1 (5.02 mg/200 mg Formula)	F2 (26.42 mg/200 mg Formula)	F3 (22.1 mg/200 mg Formula)	F6 (53.16 mg/200 mg Formula)
0	0.00 [0.0]	0.00 [0.00]	0.00 [0.00]	0.00 [0.00]
1	1.01 ± 0.01 [2.53 ± 0.03]	2.43 ± 0.22 [32.1 ± 2.86]	0.01 ± 0.001 [0.08 ± 0.001]	0.00 [0.00]
2	2.56 ± 0.44 [6.43 ± 1.09]	6.76 ± 0.12 [89.3 ± 1.61]	2.67 ± 0.06 [ 29.54 ± 0.006]	0.00 [0.00]
3	6.23 ± 0.81 [15.63 ± 2.03]	14.2 ± 2.54 [187.58 ± 34.02]	2.82 ± 0.23 [31.16 ± 2.43]	0.00 [0.00]
4	-	16.5 ± 4.59 [217.97 ± 60.64]	2.87 ± 0.61 [31.68 ± 6.75]	0.50 ± 0.01 [13.29 ± 0.27]
5	7.41 ± 1.12 [18.61 ± 2.78]	17.2 ± 3.88 [227.21 ± 49.83]	4.40 ± 0.98 [48.62 ± 10.54]	0.62 ± 0.08 [16.39 ± 2.03]
6	-		-	0.71 ± 0.08 [18.96 ± 1.99]
7	-		-	0.76 ± 0.11 [20.29 ± 2.79]
8	-		-	0.79 ± 0.21 [20.91 ± 5.45]
9	-		-	0.87 ± 0.08 [23.21 ± 1.99]
10			6.35 ± 1.05 [70.17 ± 11.42]	0.91 ± 0.02 [24.28 ± 0.61]

**Table 7 molecules-28-00748-t007:** The equation coefficients of determination (R^2^) and the “*n*” value for each mathematical model for PMZ.

The Fitted Mathematical Model	Equation	PMZ-R^2^	PMZ-*n*
Zero order	y = 26.523X + 8.248	0.8796	
First order	y = 0.332X + 0.945	0.5601	
Higuchi model	y = 28.081X − 11.319	0.9935	
Hixson Crowell model	y = −0.7478X − 2.186	0.6827	
Korsmeyer–Peppas model	y = 2.461X + 1.779	0.9934	2.46

## Data Availability

The data are contained within the article.

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
