# Peer review of "Taste Masking of Promethazine Hydrochloride Using l-Arginine Polyamide-Based Nanocapsules"

_molecules, 2023, doi:10.3390/molecules28020748_

Round 1

Reviewer 1 Report

Manuscript number molecules-2054900 entitled “Taste masking of promethazine hydrochloride using L-arginine 2 polyamide-based nanocapsules”

In this work, authors synthesized nanocapsules of polyamide based on L-arginine which were used to encapsulate Promethazine hydrochloride. Since my point of view, this work presents some drawbacks since some results are not discussed appropriately and there are some issues about the synthesis of the nanocapsules that must be addressed before publication. Therefore, this manuscript could not be accepted for publication in Molecules in its present form. Please, see below my comments.

Comments:

a)      In introduction part, authors should highlight the novelty of the present work. Has the synthesis method been reported previously for the preparation of Arg-PA?

b)      Authors should study more deeply the effect of the Organic: Aqueous medium in order to prove their hypotheses.

c)      FTIR results must be discussed more deeply. Also the description of the signals must be more precise and descriptive.

d)     The bands of PMZ in the loaded NCs could not appear due to the low amount of PMZ present. No overlapping of signal are observed since my point of view. Also, as PMZ was encapsulated, FTIR could not detect it.  

e)      In the FTIR spectrum of PMZ. Why the signal around 2300-2400 cm-1 was labeled as C-H stretching? Please clarify.

f)       RMN results must be discussed more deeply.

g)      In table 2, the data for the physical mix are missing.

h)      Why did authors measure the PMZ by DLS? Is it is possible? Is it necessary? Why? Please discuss about this in the manuscript.

i)        Has the Tg value for Arg-PAA been reported previously? Why the glass transition of the polymer was not observed in figures 5c and 5d?

j)        In table 4, what is the meaning of the values between square brackets?

Minor comments:

k)      The used template is not for Molecules

l)        What is the meaning of # in figure 7?

Reviewer 2 Report

The manuscript by Dahmash and coworkers describes the use of L-arginine polyamide-based nanocapsules to encapsulate promethazine hydrochloride. The nanocapsules show a slow release of the drug, masking the bitterness and improving the amount of fluid intake in in vivo experiments. However, the manuscript is not well presented and some experiments need to be repeated. I recommend a major revision of the manuscript.

The changes suggested are the following:

Section 2.1: it is not clear if the Arg-PA polymer has been reported in the literature previously. This should be clearly specified in the manuscript.

Figure 2: the wavelength of the peaks should be specified. Spectra c does not show the peaks of the drug. This should be revised.

Figure 3: the sharp signals of the 13C NMR may indicate that the molecular weight of the polymer is low. This needs to be analyzed. The solvent and temperature should be specified.

Figure 5: DSC curves do not show properly the peaks. The measures should be repeated using a higher temperature.

Figure 6: the XRD diffractograms do not show peaks of both polymer and encapsulated drug. They should be measured again.

Table 3: HPLC traces should be presented or provided in the suplementary material.

Section 2.9: The in vivo study is not properly described. It is not clear how is related the fluid intake with the encapsulation and release of the drug.

Section 2.10: it is not described how the release study was performed

Table 5: The plots used for the fittings should be presented in the manuscript

The conclusions of the manuscript should be improved to present all data presented in the manuscript.  

English should be revised along all the manuscript, and format mistakes such as cm-1 --> cm-1 should be fixed.

The template of the manuscript seems to be molecules but it also has Pharmaceuticals at the header and footer, please check.

Round 2

Reviewer 1 Report

Manuscript number molecules-2054900 entitled “Taste masking of promethazine hydrochloride using L-arginine 2 polyamide-based nanocapsules”

After a revision of the revised version of this manuscript, authors have addressed all my comments and improved the manuscript. Therefore, this manuscript can be accepted for publication in Molecules

Author Response

Dear Reviewer

Many thanks for your review of the manuscript. 

Reviewer 2 Report

The authors have included all the requested changes and the manuscript has improved significantly, however, some minor corrections need to be addresed. The manuscript is suitable for publication after addressing the following minor corrections:

Figure 1: the figure and caption must indicate the stereochemistry of 1,3-cyclohexane and also of L-argininge

Figure 2: spectra b) and c) look the same. If we compare a with c, the signal of HCl at 2400 cm-1 disappears, but this can be an effect of the pH. It is not clear what information the authors can obtain from these data. Please clarify this in the narrative.

Figure 3 and Table 3: the chemical structure looks weird, please fix them

Figure 5: d) plot looks very strange, the large change at ca 25 ºC seems a problem in sample measurement. This experiment should be repeated and the figure should be modified.

Figure 6: artifacts at low angles (around 0º) should be removed. In Figure d) there are some peaks of crystalline Arg-PA, this should be explained in the narrative.

English should be revised throughout the manuscript, to fix errors and non-standard sentences. For example in line 185:

1H NMR and 13C-NMR characterised the synthesised Arg-PA polymer. --> 

The synthesized Arg-PA polymer was characterized by 1H NMR and 13C-NMR spectroscopy.

Author Response

Dear Reviewer 
